# High Presence of NETotic Cells and Neutrophil Extracellular Traps in Vaginal Discharges of Women with Vaginitis: An Exploratory Study

**DOI:** 10.3390/cells11203185

**Published:** 2022-10-11

**Authors:** Fabiola Zambrano, Angélica Melo, Rodrigo Rivera-Concha, Mabel Schulz, Pamela Uribe, Flery Fonseca-Salamanca, Ximena Ossa, Anja Taubert, Carlos Hermosilla, Raúl Sánchez

**Affiliations:** 1Center of Excellence in Translational Medicine—Scientific and Technological Bioresource Nucleus (CEMT—BIOREN), Faculty of Medicine, Universidad de La Frontera, Temuco 4780000, Chile; 2Department of Preclinical Sciences, Faculty of Medicine, Universidad de La Frontera, Temuco 4780000, Chile; 3Department Pathological Anatomy, Faculty of Medicine, Universidad de La Frontera, Temuco 4780000, Chile; 4Ph.D. Program in Medical Sciences, Faculty of Medicine, Universidad de La Frontera, Temuco 4780000, Chile; 5Department of Internal Medicine, Faculty of Medicine, Universidad de La Frontera, Temuco 4780000, Chile; 6Department of Public Health, Centro de Excelencia Capacitación, Investigación y Gestión para la Salud Basada en Evidencias (CIGES), Universidad de La Frontera, Temuco 4780000, Chile; 7Institute of Parasitology, Justus Liebig University Giessen, 35390 Giessen, Germany

**Keywords:** neutrophil extracellular traps, *Candida albicans*, *Trichomonas vaginalis*, bacterial vaginosis, infectious vaginitis

## Abstract

Infectious vaginitis is a microbiological syndrome of great importance in public health that affects millions of women worldwide. However, no studies have explored the phenomenon of the production of the neutrophil extracellular traps (NETs) that are released into the female reproductive tract in these pathologies. This study aimed to determine the presence of NETosis in vaginal discharges of women with bacterial vaginosis, candidiasis, and trichomoniasis by characterizing NETs. Extracellular DNA with neutrophil elastase and citrullinated histones was identified to confirm the NET components (*n* = 10). The concentration, phenotypes of NETs, and number of NETotic cells were determined. The results showed an increase in NETotic cells in women with *Candida albicans* (CA) and *Trichomonas vaginalis* (TV) and an increase in NETs in TV-induced vaginitis. Samples of CA- and TV-infected women showed different NET phenotypes (*diff*NETs, *spr*NETs, and *agg*NETs); *diff*NETs were found in high concentrations in samples with CA and were increased in three types of NETs in TV infections. Samples with intermediate microbiota and bacterial vaginosis showed increased NETotic cells while the intermediate microbiota presented a higher concentration of NETs. Therefore, alterations in the microbiota and the presence of fungal and parasitic infections are important stimuli for the activation and induction of NETosis, and their cytotoxic effects could enhance tissue damage.

## 1. Introduction

Infectious vaginitis is a microbiological syndrome of great importance to public health because it affects millions of women worldwide [1]. The most frequent is bacterial vaginosis (BV), a vaginal dysbiosis associated with several bacteria [2] that has a global prevalence of over 30% in women of reproductive age [3]. Vulvovaginal candidiasis (VVC) is the second most common cause of infectious vaginitis after BV, with an incidence between 12.1% and 57.3% in symptomatic women; *Candida albicans* (CA) is the most common causative agent of VVC [4]. In addition to bacterial and fungal pathogens, trichomoniasis, the most common non-viral sexually transmitted infection in the world [5], is caused by the flagellated protozoan parasite *Trichomonas vaginalis* (TV). TV infects 173 million people each year worldwide; 32 million of these infections occur in sub-Saharan Africa [6,7] and there is a prevalence of 3.9% according to epidemiological data in Latin America [8].

Polymorphonuclear neutrophil (PMN) recruitment and tissue infiltration to sites of infection represent the first line of defense within the host innate immune system against invasive agents [9]. In vitro studies have reported variable concentrations of PMN during vaginal infections with CA [10], BV [11], and TV [12].

CA causes tissue damage in vitro such as edema, vacuolization, and separation of keratinocytes [13,14,15,16]. The PMN arsenal of effector functions include phagocytosis, degranulation, and the production of reactive oxygen species (ROS) [17]. Extracellular chromatin structures released by PMN in the form of a fine network known as neutrophil extracellular traps (NETs) are of special interest; these are at the forefront of the renewed interest in the biology of PMN. The release of NETs from PMN is of particular importance in trapping and killing fungal filaments [18]. Therefore, this could be a key defense mechanism in the physiopathology and outcome of vaginal infections, as demonstrated in male urogenital infections in which the presence of bacteria such as *Ureaplasma urealyticum, Chlamydia trachomatis,* and *E. coli* had a negative impact on sperm quality that was associated with an increase in extracellular traps. [19].

The molecular process of NET release, also known as NETosis, is characterized by an initial oxidative burst activity through NADPH oxidase and citrullination of nuclear histones by peptidylarginine deiminase 4 (PAD4) [20]. In this post-translational modification, arginine residues are transformed into citrulline and the highly condensed chromatin decondense, resulting in the expansion of the PMN nucleus and the rupture of the nuclear membrane [21]. The PMN lose membrane integrity, which culminates in the release of NETs into the extracellular space [22]. NETs are fibrous and sticky extracellular DNA structures studded with histones (H1, H2A/H2B, H3, and H4) and other granular components such as neutrophil elastase (NE), myeloperoxidase (MPO), cathepsin G, lactoferrin, metalloproteinase-9, pentraxin, and cathelicidin, among others [23]. NETs are very effective antimicrobial structures that are capable of trapping and eliminating pathogenic agents such as bacteria, fungi, protozoans, and large parasites. They can perform their function in different phenotypes such as diffuse NETs (*diff*NETs), spread NETs (*spr*NETs), and aggregated NETs (*agg*NETs) [20]. Different in vitro and in vivo studies have elucidated the underlying signaling pathways and have highlighted the biological importance of NETosis against extracellular pathogens [24,25]. Extracellular traps interacted with TV [26] in a mouse model, but this was not a suitable host for TV. Understanding the role of human NETs against TV in vivo requires evaluating whether these in vitro observations have physiopathological relevance. In relation to CA, most in vivo NET-related studies have focused on the lung or vasculature [27,28,29]. However, these are not the target organs of CA-induced vaginitis in humans [20]. The aim of this study was to determine the presence of NETotic cells, NETosis, and NET-derived mechanisms in vaginal discharge samples of women diagnosed with BV, CA, and TV. We aimed to characterize and quantify NETs and their different phenotypes associated with the diagnosis of these pathologies in samples as close as possible to the in vivo conditions.

## 2. Materials and Methods

### 2.1. Study Population and Ethical Approval

The samples were obtained from women (*n* = 10) between the ages of 18 and 51 years who were in the follicular phase of the ovarian cycle through obstetric consultations at different primary health care services in Temuco, Chile. The research protocol was authorized by the Scientific Ethics Committee of the Universidad de La Frontera (file #045-2017). All volunteers participating in this study signed an informed consent.

The inclusion criteria were: women over 18 years of age who consulted spontaneously due to an abnormal vaginal discharge or who were clinically diagnosed at the time of the routine consultation in the examination. The exclusion criteria were: women under 18 years of age, women with antibiotic treatment during the last 30 days and/or who were taking immunosuppressive drugs or who presented immunosuppression, women in their menstrual period at the time of sample collection, and women who had sexual intercourse in the last 48 h.

All the women included in this study were symptomatic (except those in the control group) and presented an abnormal vaginal discharge and the presence of erythema on the vaginal walls and external genitalia, in addition to itching.

Sample eligibility criteria: within the universe of 125 archival samples, 10 samples were chosen for this study that met the eligibility criteria; these were: slices containing samples of TV+ and CA− (*n* = 3), samples of CA+ and TV− (*n* = 3), and samples from the control group that were negative for CA and TV with normal vaginal microbiota according to Nugent’s criteria (*n* = 3). Faced with a second scenario, we proposed to analyze the samples from the point of view of the microbiota; for this, an additional sample of TV− and CA− with bacterial vaginosis was incorporated in order to have a minimum number of samples to perform the quantitative analyses.

The control-group samples came from archival samples from women who attended a routine consultation for Pap smear collection and who presented clinically normal flows (physiological flows), had no manifestations in the external genitalia (e.g., erythema, edema, among others), and had no abnormal signs on speculoscopy. They corresponded to samples from healthy patients without an abnormal vaginal discharge and with normal microbiota according to Nugent’s criteria classification. Controls were defined as CA− and TV− negative (determined via PCR) and normal microbiota determined by Gram staining and by applying Nugent’s criteria for microbiota classification.

### 2.2. Sample Collection and Preparation

Speculoscopy was used to obtain two samples with a sterile cotton swab from the posterior fornix during a routine gynecological examination. Two vaginal smears were immediately placed on sterile glass slides and dried at room temperature (RT). The swabs were placed in sterile phosphate-buffered saline (PBS) 1× (Sigma-Aldrich, St. Louis, MI, USA) and all samples were immediately transported to the Molecular Immunoparasitology Laboratory at the Universidad de La Frontera, Temuco, Chile, within 24 h of sample collection for further investigation.

### 2.3. Etiological Diagnosis of Vaginal Discharges Using PCR

DNA was isolated from the swab samples using the E.Z.N.A.^®^ Tissue DNA kit and stored at −80 °C until processing. Polymerase chain reaction (PCR) was performed with the primers GH20/PCO4 specific to the β-globin gene (internal control). Genomic DNA served as the positive control. The detection of *T. vaginalis* (TV) and *C. albicans* (CA) was via: (1) clinical signs, (2) microscopic examination, and (3) the presence of CA and TV confirmed by conventional simple PCR with previously published primers [30] and sequences [31], respectively. For the positive controls, clinical samples positive for TV and CA were used that were obtained through fresh sampling for TV and from cultured samples for CA. Amplified bands were visualized in a 1.6% agarose gel using a UV transilluminator. A sample was considered positive when a band of 300 bp (pb) for *T. vaginalis* or 496 bp for *C. albicans* was observed.

### 2.4. Gram Staining, Nugent’s Criteria, and Presence of PMN

Gram staining was performed using the Gram Staining Kit (Becton Dickinson, Franklin Lakes, NJ, USA). Briefly, the samples on the slides were dyed with gentian violet for 1 min, washed with water, and treated with lugol (fixative of gentian violet) for 1 min. Next, the samples were washed and an alcohol–acetone mixture was added for ±10 s, washed, and then stained with safranin. Finally, the samples were washed and dried at RT, covered with mounting medium (Entellan-Merck, Rahway, NJ, USA), and a coverslip was placed on top. Using a TissueFAXS i Plus cytometer (TissueGnostics, Vienna, Austria), the Gram-positive bacteria were visualized as purple, and the Gram-negative bacteria were visualized as red.

The reading was interpreted by applying Nugent’s criteria [32], which classified the microbiota as normal (NM) (0–3), intermediate (IM) (4–6), or bacterial vaginosis (BV) (7–10). The count was performed in 20 fields with a 100× objective; Gram-positive bacilli (*Lactobacillus* spp.), Gram-variable/Gram-negative coccobacilli (*Gardnerella vaginalis*/*Prevotella* spp.), and curved Gram-variable bacilli (*Mobiluncus* spp.) were quantified by assigning one point per field. A normal microbiota had a predominance of *Lactobacillus* spp.; an intermediate microbiota showed Gram-positive bacilli and Gram-variable/Gram-negative coccobacilli; and in bacterial vaginosis, the presence of *Lactobacillus* spp. was almost nonexistent, with >30 Gram-negative/Gram-variable bacterial morphotypes per observation field. The presence of *T. vaginalis* trophozoites, blastoconidia, and pseudohyphae was also recorded.

The presence of PMN was detected and semi-quantified using the following criteria: scarce (+, 0–5 cells), moderate (++, 5–10 cells), and abundant (+++, >10 cells). We assessed 20 fields in homogenous areas using the 100X objective with immersion oil. 

### 2.5. Identification of Extracellular DNA and Neutrophil Elastase (NE) in Vaginal Discharges

An immunofluorescence microscopy analysis was conducted on the vaginal smears on slides using the following methodology. The samples were blocked with 20% bovine serum albumin (BSA; Sigma-Aldrich) and 0.005% saponin in sterile PBS for 30 min at RT. Two washes were performed with 500 μL of 1× PBS. Following washing, the primary antibody was added to detect NE (1:300 dilution, anti-rabbit ab 68672, Abcam, Cambridge, UK) and incubated for 1 h at RT. Next, the samples were washed twice with 500 μL of 1× PBS and the secondary antibody was added (Alexa Fluor 488 anti-rabbit, reference no. A11008; Life Technologies, Carlsbad, CA, USA).

The samples were incubated for 1 h at RT in the dark. The samples were washed twice with 500 μL of 1× PBS and incubated with Sytox Orange^®^ (staining for DNA, 1:2000 in PBS 1×) for 15 min. After two washes, an assembly medium was added (REF:00-4959-52, Invitrogen) and each sample was covered with a coverslip for subsequent evaluation using a TissueFAXS i Plus cytometer (TissueGnostics, Vienna, Austria).

### 2.6. Nuclear Staining and Citrullinated Histone Identification (H4cit3)

The samples were blocked with 20% BSA and 0.005% saponin in sterile PBS for 30 min at RT, stained with anti-H4cit3 (anti-rabbit polyclonal, 07-596, Merck Millipore, Burlington, MA, USA), and incubated at RT for 1 h. Later, the samples were washed with 500 μL of 1× PBS and the secondary antibody was added (Alexa Fluor 405 goat anti-rabbit IgG; ref. A31556, Invitrogen). After 1 h of incubation, the samples were washed and incubated with Sytox Orange^®^ for 15 min. After two washes, a mounting medium was added and each sample was covered with a coverslip for subsequent evaluation using a TissueFAXS i Plus cytometer (TissueGnostics, Vienna, Austria).

To perform the nuclear analysis of PMN, five images were taken at random (four corners and one in the middle, avoiding bubbles) for each sample at 20× magnification and 1× zoom using the channel for Sytox Orange^®^ staining (red stain). Images of each sample were analyzed using the ImageJ^®^ editing program as described previously [21]. Briefly, images were transformed to 8-bit and the data corresponding to the area of each particle were collected in pixels using the “analyze particles” function (individual analysis of each cell). The data were then transformed to area units (μm^2^) and the results were analyzed using GraphPad, Prism 9.0.0, https://www.graphstats.net/graphpad-prism, (accessed on 24 October 2020). PMN with an area >80 μm^2^ were considered as NETotic cells [21].

### 2.7. Identification and Quantification of Different NET Phenotypes

To unveil the different NET phenotypes, immunofluorescence of H4cit3 and nuclear staining were performed as described previously. The samples were analyzed using an Olympus IX81 inverted fluorescence microscope equipped with an Olympus XM10 digital camera. The *diff*NETs, *spr*NETs, and *agg*NETs were analyzed microscopically using five randomly selected images for each group and classified morphologically according to previous reports [33,34]: (I) *diff*NET were composed of a decondensed extracellular chromatin network stained with compact globular antimicrobial proteins that were 15−20 µm in diameter; (II) *spr*NETs consisted of structures in the form of smooth and extended bands of decondensed chromatin with antimicrobial proteins composed of fine fibers that were 15−17 μm in diameter; and (III) *agg*NETs were composed of a very high density of NETs released from numerous PMN simultaneously forming large DNA-derived aggregates covered with global histones and antimicrobial components [35]. In each sample, all phenotypes were counted and compared with that in the control group without infection. Data are expressed as the average of *diff*NETs, *spr*NETs, and *agg*NETs obtained in each group.

### 2.8. Statistical Analysis

The results are presented as the mean ± standard deviation (SD). Prism 9.0.0 software (GraphPad^®^) was used for statistical evaluation. D’Agostino–Pearson’s K2 test was used to assess the Gaussian distribution; when the numerical results did not pass the test of normality, they were transformed to a logarithmic scale. One-way analysis of variance (ANOVA) was used with a subsequent Dunnett’s or Bonferroni test. The statistical significance was set at *p* < 0.05.

## 3. Results

### 3.1. Characterization of Vaginal Discharges and Identification of NETs Ex Vivo

Representative images of vaginal PCR tests positive for CA (490 bp) and TV (300 bp) are shown in Figure 1A. Figure 1B shows PCR positive or negative results for CA and TV, or negative results for both infectious agents that were included in the control group. According to Nugent’s criteria, positive CA samples presented characteristics of intermediate microbiota (2/3) and bacterial vaginosis (1/3), all of which were negative for TV. Likewise, positive TV samples presented characteristics of intermediate microbiota (2/3) and bacterial vaginosis (1/3), all of which were negative for TV. PMN identification was not directly proportional to the type of pathogenic agent or microbiota in each sample. An abundant number of PMN was visualized in vaginal discharges: 1/3 in samples with CA and 2/3 in samples with TV. The control group (normal microbiota, negative CA and TV) presented a scarce-to-moderate number of PMN (Figure 1C). Representative images of epifluorescence microscopy showing neutrophil infiltration in vaginal discharges for CA, TV, and the control group are shown in Figure 1D.

NETs were present in all vaginal samples with CA- and TV-induced vaginitis (Figure 1D) as well as in the control group. Labeling of H4cit3 attached to extracellular DNA confirmed the process of NETosis (Figure 1D). In addition, the visualization of NE on the released extracellular chromatin structures corroborated the presence of NETs (Figure 2e,f,h,i,k). The samples diagnosed via PCR as CA infections presented pseudohyphae that were easily visualized using Gram staining (Figure 2b,c). In the samples obtained from patients diagnosed with TV-induced vaginitis, the extracellular parasite stages (i.e., trophozoites) were visualized with Gram staining (Figure 2d). Different visual findings in the samples with CA- and TV-induced vaginitis indicated the release of NET-derived strands from PMN that were connected to adjacent cells (Figure 2f) and cells completing NETosis with total loss of DNA at the intracellular level (Figure 2).

### 3.2. Quantification of NETotic Cells in Vaginal Discharges

The nuclear PMN expansion analyses showed an average area of 113.9 μm^2^ in CA infections (*n* = 796) and 138.4 μm^2^ in TV infections (*n* = 1646); both groups presented a greater nuclear expansion (*** *p* < 0.001) than the control group (66.1 μm^2^, *n* = 786) (Figure 3A). In addition, when grouping the samples by type of microbiota according to Nugent’s criteria, the group with normal microbiota (*n* = 786) presented a nuclear expansion of 66.1 μm^2^, which was significantly smaller (*** *p* < 0.001) than that of the group with intermediate microbiota (125.1 μm^2^, *n* = 1757) and the group with bacterial vaginosis (114.2 μm^2^, *n* = 1022) (Figure 3B). The percentage of PMN with a nuclear area greater than 80 μm^2^ was higher (*** *p* < 0.001) in the CA (81.2 %, *n =* 796) and TV (68.0%, *n =* 1646) groups than that in the control group (19.1%, *n* = 786) (Figure 3C). When the samples were classified by type of microbiota, the groups that presented with intermediate microbiota and bacterial vaginosis showed a high percentage of PMN with nuclear areas greater than 80 μm^2^ (*** *p* < 0.001) compared to that in the group with normal microbiota (76.4%, *n* = 1757; 64.7%, *n* = 1022; 19.1%, *n =* 786, respectively) (Figure 3D). The microscopic images in Figure 3 show nuclear expansion using Sytox Orange staining (red fluorescence).

### 3.3. Identification and Quantification of Different NET Phenotypes

The identification of different NET phenotypes in vaginal discharges showed the presence of *diff*NETs, *spr*NETs, and *agg*NETs in all samples and in control groups via identification of H4Cit3 attached to the extracellular DNA structures. Representative images are shown in Figure 4. Quantification of different NET structures (Figure 5) revealed a significant increase in the number of *spr*NETs in the samples originating from TV-induced vaginitis (*** *p* < 0.001). However, the CA-induced vaginitis group did not show any difference compared to that in the control group. Quantification of *diff*NETs showed a significant increase in the groups with CA- (* *p* < 0.05) and TV-mediated vaginitis (*** *p* < 0.001) compared to that in the control. The *agg*NETs were significantly increased in the TV group (* *p* < 0.05) compared to those in the control group. The CA-mediated vaginosis group showed no differences in this parameter (Figure 5A). When grouping the results by type of microbiota, quantification of NETs by focusing on *spr*NETs and *agg*NETs showed a significant increase in *diff*NETs in the group with intermediate microbiota (** *p* < 0.01) compared to that in the control; there were fewer *diff*NETs in the group with BV compared to those in the control group. In addition, the group with bacterial vaginosis had a lower number of *diff*NETs (* *p* < 0.05) than the group with normal microbiota (please refer to Figure 5B).

The quantification of different phenotypes of NETs showed a highly significant increase in these structures (i.e., *diff*NETs) in the TV vaginitis group (*** *p* < 0.001) compared to those in the control. In contrast, the CA group showed no significant differences (Figure 5C). When grouping by microbiota type, the global quantification of NETs increased in the group with intermediate microbiota (*** *p* < 0.001). The group with bacterial vaginosis showed no differences when compared to the control group (Figure 5D).

## 4. Discussion

PMN are the most abundant innate immune cells and the first line of defense against many pathogenic microorganisms, including the human pathogenic fungus CA and the sexually transmitted parasitic protozoa TV. CA and TV are biological stimuli for NET formation in vitro [5,20]. This study explored the presence of NETotic cells, NETs, and different phenotypes of NETs and their association with candidiasis, trichomoniasis, and bacterial vaginosis in vaginal smears from women with a clinical diagnosis of vaginitis.

Vaginal discharges contain a variable amount of PMN accompanying vaginitis and display numerous effector mechanisms against pathogenic microbes. PMN-recruited professional phagocytes empty their granule content, releasing peptides/proteins such as elastase, MMP9, pentraxin, MPO, lactoferrin, cathepsin, and peroxidase, as described previously, during the degranulation process [22]. NETs have a large amount of these enzymes, which are released into the extracellular space along with other antimicrobial proteins, DNA, and histones (e.g., H2A) that originate mainly from the PMN nucleus [22]. This results in an increase in extracellular elastase in the vaginal discharge, which could be explained by the high presence of NETotic cells ex vivo, most likely triggered by pathogens. An increase in both NETotic cells and NET formation (suicidal NETosis) was detected in the vaginal discharges of women suffering from acute CA vaginitis, which was consistent with earlier reports indicating a strong activation of PMN following exposure to different CA stages in vitro [20,36]. Exposure of PMN to CA yeast cells results in significant phagocytosis; however, exposure to hyphae leads to NETs extrusion, as these fungal stages appear to be too large to be eliminated intracellularly [37]. In such cases, the release of NETs has emerged as the predominant strategy of PMN to trap and kill large pathogens, including CA hyphae [38] and various large helminth species [34]. The development of NETosis in samples from CA patients showed varying degrees of inflammatory development. A higher number of NETotic cells was observed in CA-induced vaginitis when compared to that in clinical cases of TV-induced vaginitis. This phenomenon could be explained by the presence of blastoconidia and pseudohyphae, which might have stimulated a greater number of PMN to initiate NETosis. NETotic cells were present in large numbers in the vaginal discharges of women diagnosed with trichomoniasis; in addition, there was an increase in the different NET phenotypes. The formation of TV-induced NETs is related to the phosphorylation of P38 and ERK1/2 MAPK signaling pathways in vitro [26]. PMN quickly kill TV through trogocytosis but not independently of NETs [39]. In addition, both live and dead TV can induce NET structures [26]. With respect to the total NETs, the greatest number of total NETs and a smaller proportion of NETotic cells were observed in the TV group, indicating that several PMN had possibly already extruded DNA into the extracellular surroundings at the time of sample evaluation. TV trophozoites are powerful inducers of NETosis in mouse models; however, very little is known about TV-derived antigens, secreted proteins, or parasite-derived molecules involved in NET release in humans.

Since the discovery of NETs by Brinkmann et al. in 2004, various NET phenotypes have been reported, including cell-free NETs, anchored NETs, *spr*NETs, *diff*NETs, and *agg*NETs. Invasive protozoan and metazoan parasites can induce all five NET types [39], with each exhibiting different functions [22]. Patients with clinically manifested CA and TV vaginitis presented with at least three different morphological phenotypes (*diff*NETs, *agg*NETs, and *spr*NETs), with a higher presence of *agg*NET in the case of TV-mediated vaginitis. These *agg*NETs can trap large and highly motile nematodes, large quantities of motile protozoan parasites [40], or even highly motile human spermatozoa in vitro [41]; this suggests not only size sensing, but also motility sensing by human PMN. These different NET phenotypes were confirmed through the co-localization of enriched extracellular structures with DNA such as the association of large amounts of H4Cit3 histones [19]. The role of histone citrullination in the development of CA-induced NETosis and that of chromatin decondensation accompanied by nuclear expansion were described in in vitro studies [20]. The physiological stimuli produced by fungi and crystals induced histone citrullination during NETosis [42]. This role of histone citrullination in NETosis should be elucidated further because abnormal citrullination can lead to the development of autoimmune disorders and cancer in vivo [43].

In CA-induced vaginitis, the yeast form is relevant for dissemination but the growth of CA hyphae is essential for tissue invasion and destruction [44]. In this CA-mediated tissue damage, the presence of abundant NETs as extracellular traps (ETs) can be cytotoxic to adjacent cells as well [44]. Although the high presence of NETotic cells contributes to the immunity against TV infections, the critical protective role of PMN becomes problematic in the case of anti-fungal defense mechanisms [20]. In trichomoniasis, the vaginal squamous epithelium is the main site of infection, although the parasite can reach the urethra and endocervix [26]. The arsenal of PMN can damage the host’s tissues; therefore, their function is strictly regulated through three main strategies: phagocytosis, degranulation, and release of NETs [42]. NETs do have a specialized immunoprotective function; however, the list of conditions in which non-digested NETs result in pathology is continuously expanding [42]. Therefore, instances in which NETs directly kill or damage the epithelium and/or endothelium are abundant [45]. Excessive NETosis damages the epithelium in CA pulmonary fungal infection [28] and the endothelial lining in transfusion-related acute pulmonary injury [42,46]. In addition, in sepsis and acute injury, free-circulating histones are cytotoxic due to their ability to compromise the integrity of the cell membrane [47]. Histones bonded to NETs play a central role in the cytotoxicity mediated by these structures [42,47]. The ability of ETs to damage tissues explains why it is likely that their release is limited to infections that cannot be eliminated using less harmful strategies [42].

In the case of TV and CA vaginitis infections, which are associated with altered microbiota, future evaluations of IM and BV might offer relevant information. Bacterial vaginosis (and intermediate microbiota, the stage prior to BV) is considered a polymicrobial disease in which *Gardnerella* spp. and *Atopobium vaginae* are the main agents [2]. Beginning with the first study by Brinkmann et al. (2004), NETs were described as potent effectors against Gram-positive and Gram-negative bacteria [22]. Previous studies have shown that the presence of PMN in BV is still under debate [48,49]. Nevertheless, as-investigated alterations to the microbiota should be considered as stimuli in the initiation of PMN recruitment, activation, and induction of NETosis. According to our observations, it would be possible to find a high percentage of NETosis in IM and BV. In addition, using the number of total NETs, it might be possible to observe more of them, with a predominance of *diff*NETs in the vaginal discharges of women with IM.

The abundance of NETotic cells and NET formation in vaginal discharges with altered microbiota, CA infections, and TV infections could possibly be linked to female infertility disorders, as demonstrated for infections of the male genital tract [19]. There is a high frequency of BV in women with infertility [50,51]. In the case of trichomoniasis, its impact is not limited to vaginal or urethral infections, but is extended to HIV transmission and acquisition [6,48] and to the risks of cervical cancer [52] and prostate cancer [53]. TV is also related to adverse pregnancy outcomes [52] and male and female infertility [54,55,56,57]. This link between the presence of NETs and infertility was corroborated by earlier reports in which we described the detrimental effects of these structures on sperm function [19,25,57]. Therefore, the deposition of spermatozoa, either via coitus or artificial insemination, into vaginal microenvironments altered due to pathogenic agents might augment the presence of ETs that modulate not only local innate immune responses, but also hamper sperm functions. Particularly in human cases of idiopathic infertility, excessive NETosis induction should be considered a possible etiology, as postulated earlier [25].

In conclusion, the impact of trichomoniasis, BV, and candidiasis on public health requires constant research into the pathogenic mechanisms and early host innate immune responses involved in these gynecological infections. Elucidating the ability of NETs to trap microorganisms is critical to a better understanding of their role in the physiopathology of many infections; however, their pathogenic potential has garnered attention due to the cytotoxic effects of NETotic cells and NETs, which can enhance tissue damage. The alterations in the vaginal microbiota and the presence of pathogens appear to a significant stimuli for the initiation of PMN recruitment into the vagina, induction of the NETotic process, and the final release of NETs. A significant increase in bacterial vaginosis-induced NETs in the female reproductive tract could influence cases of idiopathic infertility. Considering that this was the first study to research the presence of NETotic cells and NETs in vaginal discharges of women with vaginitis, further studies with a larger sample size are required to support these findings.

## 5. Study Limitations

The limitation of our study was that it included a rather small sample size—only patients with clinically manifested vaginitis—and therefore perhaps was not truly representative of the entire population of women exposed to these pathogens with subclinical and/or underdiagnosed vaginitis. Therefore, it would of use to increase the number of patients for each etiological condition in future studies to verify the pivotal role of NETosis in human vaginitis.

## Figures and Tables

**Figure 1 cells-11-03185-f001:**
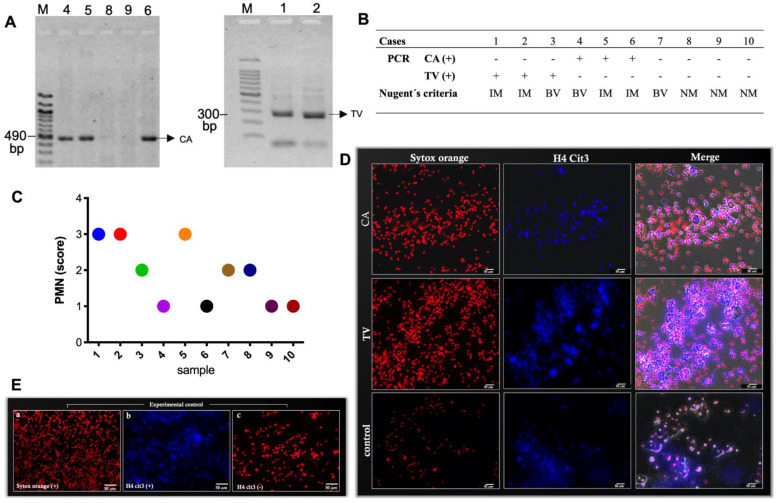
Diagnosis and characterization of vaginal discharges. (**A**) Representative images of the PCR results for the samples of vaginal discharges with CA and TV. (**B**) The results of each case in the PCR diagnosis and the classification according to Nugent’s criteria (NM, normal microbiota; IM, intermediate microbiota; BV, bacterial vaginosis). (**C**) The score for the presence of neutrophils in each vaginal discharge (pathological and controls), with 1 indicating scarce, 2 indicating moderate, and 3 indicating abundant. (**D**) Representative images of epifluorescence microscopy showing neutrophil infiltration in vaginal discharges for CA, TV, and control group. Red indicates DNA (Sytox Orange) and blue indicates H4cit3; the merge was done in a bright field. Images in (**E**) show experimental controls: (**a**) Sytox-orange-positive cells, (**b**) H4cit3-positive cells, and (**c**) H4cit3-negative cells (without primary antibody). The images were taken at 20× magnification.

**Figure 2 cells-11-03185-f002:**
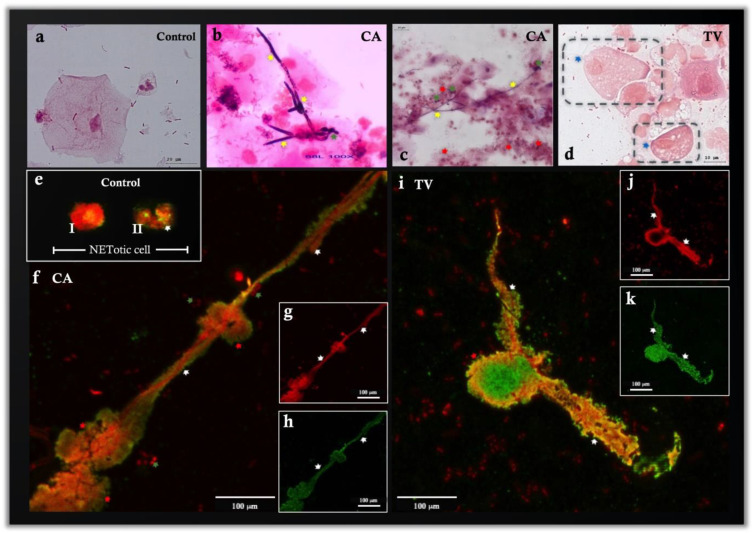
Representative images of pathological vaginal discharges. (**a**) Control group; (**b**,**c**) Gram staining of vaginal discharges with the presence of blastoconidia and pseudohyphae (corresponding to CA diagnosis using PCR). The dark purple stain shows pseudohyphae at 100× magnification. (**d**) TV in Gram staining. (**e**) Immunofluorescence of pathological vaginal discharges with neutrophils releasing NETs. Neutrophils activated by NETosis are shown in the control group with differing degrees of activation: I, compact nucleus; II, nucleus beginning to decondense. (**f**,**i**) neutrophils releasing NETs at a greater magnification (100× magnification, 100% zoom). (**g**,**j**) Extracellular DNA in red (Sytox Orange). (**h**,**k**) Neutrophil elastase in green (Alexa fluor 488). Yellow arrows indicate pseudohyphae, green arrows indicate blastoconidia, red arrows indicate PMN, white arrows indicate NETs, blue arrows indicate flagella of TV, and the outline in black indicates TV.

**Figure 3 cells-11-03185-f003:**
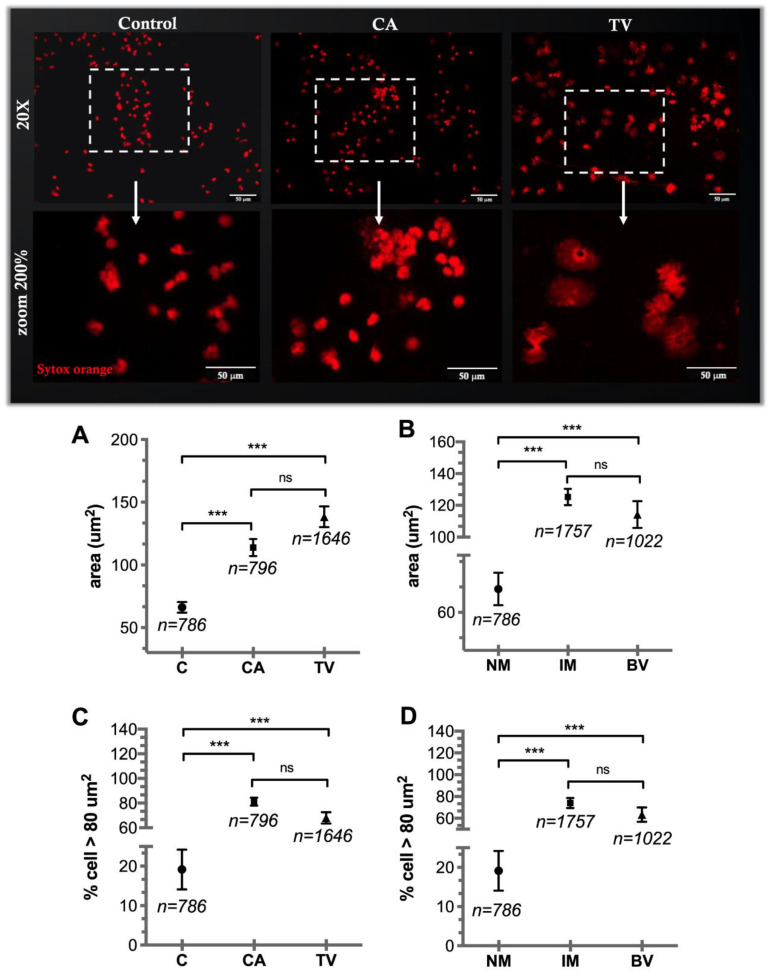
Nuclear PMN analysis of pathological vaginal discharges. The microscopic images show nuclear staining with Sytox Orange (red fluorescence) at 20× and 200× magnification. Graphs in (**A**) and (**B**) show average sizes of the cells analyzed in µm^2^ grouped into discharges diagnosed as CA and TV and classified according to characteristics of the microbiota, respectively. Graphs in (**C**,**D**) show percentages of cells with nuclei greater than 80 µm^2^ grouped according to pathology and microbiota, respectively. The data are indicated as mean ± SD; the total number of analyzed cells in each column is indicated (*** *p* < 0.001).

**Figure 4 cells-11-03185-f004:**
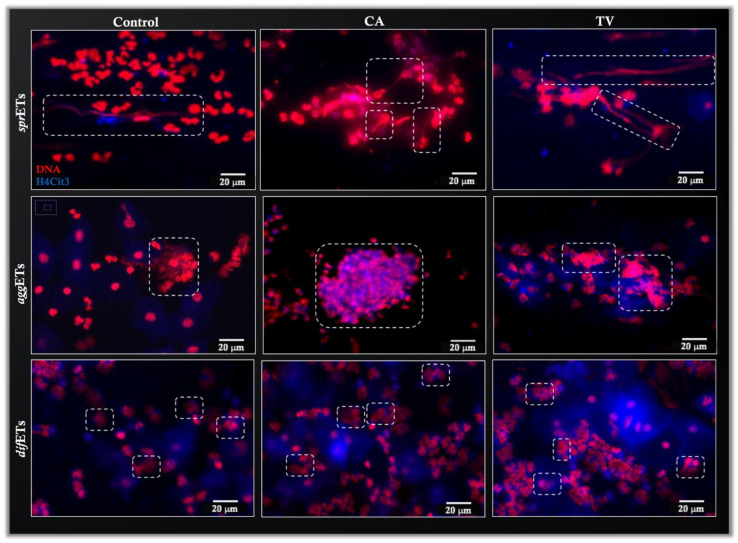
Representative images of CA- and TV-positive vaginal discharges and control (c) presenting NETs in their different morphologies. In all the images, red indicates DNA stained with Sytox Orange and blue indicates stained citrullinated histone (H4Cit3). The outline in white indicates the type of NET that each image represents: *spr*NETs, “spread”; *agg*NET, “aggregated”; *diff*NETs, “diffuse”.

**Figure 5 cells-11-03185-f005:**
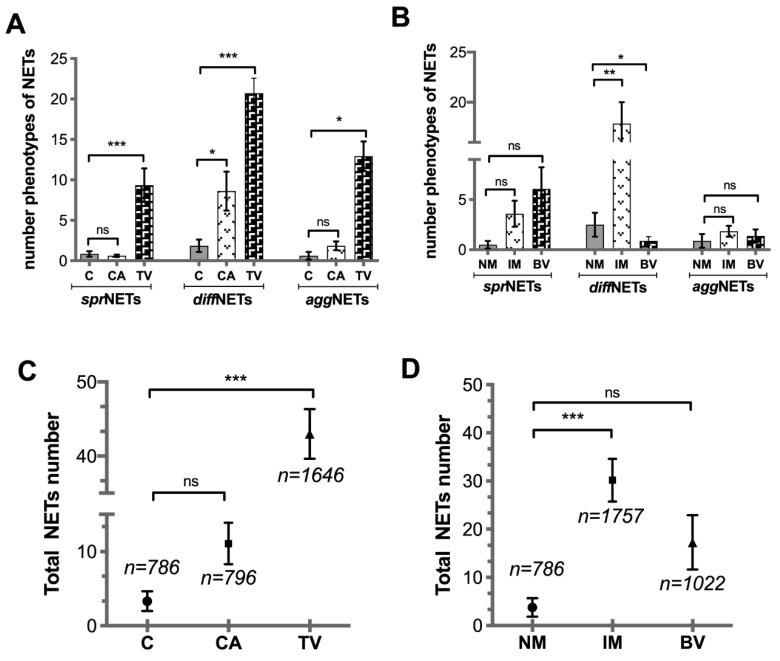
Quantification of the different NET structures. The graph in (**A**) shows types of NETs in pathological vaginal discharges classified according to the infection (CA and TV) and control (**C**); the graph in (**B**) shows the quantification of different types of NETs in groups classified according to microbiota (MN, IM, and BV). The graphs in (**C**,**D**) show the total NETs in groups classified according to the type of infection and types of microbiota, respectively. The data are indicated as mean ± SD (* *p* < 0.05, ** *p* < 0.01, *** *p* < 0.001).

## Data Availability

The data that support the findings of this study are available from the corresponding author upon reasonable request.

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
