# Peer review of "High Presence of NETotic Cells and Neutrophil Extracellular Traps in Vaginal Discharges of Women with Vaginitis: An Exploratory Study"

_cells, 2022, doi:10.3390/cells11203185_

Round 1
Reviewer 1 Report (New Reviewer)
Thank you for the opportunity to review this paper. I found it very interesting and well articulated.
Only minor comments for improvement
Abstract: consider defining NETs and NETosis in the intro
Intro:
Line 63 could be further explained
Methods:
Was the trial registered prospectively on a clinical trials registry
You mention that 10 women were chosen – what were the eligibility criteria? Were these women symptomatic for vaginitis? Please include more details on the inclusion/exclusion criteria in the methods and also more details in the results about these women. What symptoms did they have, did they have discharges? Had they been treated with antibiotics recently?
Please also include more details about the control group? Are these lab controls or women with no aetiological diagnosis? Please define in the methods how controls are defined.
Author Response
Comments from Reviewer 1:
Thank you for the opportunity to review this paper. I found it very interesting and well-articulated. Only minor comments for improvement.
Q1: Abstract: consider defining NETs and NETosis in the intro.
A1: Thanks for the comment. We have defined NETs in the abstract and introduction. The mechanism of "NETosis" is explained in the introduction section, this word is not an abbreviation, so it is not defined, but rather the concept is explained.
Q2: Intro: Line 63 could be further explained.
A2: Thanks for the feedback. We have added the requested information.
Q3: Methods: Was the trial registered prospectively on a clinical trials registry.
A3: Thanks for the comment. Our purpose was to describe the NETs’ presence in women with vaginal infections. We called “exploratory” because is an observational study and we do not do any interventions. Hence, it is a first report about the different kinds of NETs that appears in vaginal smears of women, and, of course, open a new line of investigation that may include a clinical trial in the near-future.
Q4: You mention that 10 women were chosen – what were the eligibility criteria? Were these women symptomatic for vaginitis? Please include more details on the inclusion/exclusion criteria in the methods and also more details in the results about these women. What symptoms did they have, did they have discharges? Had they been treated with antibiotics recently?.
A4: Thank you for these interesting observations. Regarding the eligibility criteria of the samples, Within the universe of 125 samples, 10 samples that met the eligibility criteria were chosen for this study. The selection criteria were: 3 plates containing samples of TV+ and CA-, 3 plates containing samples of CA+ and TV- and 3 plates of control group of negative samples for CA and TV, which also met the criteria of having normal vaginal microbiota according to criteria of Nugent. Faced with a second scenario, it was proposed to analyze the samples from the point of view of the microbiota and for this an additional sample TV-, CA- with bacterial vaginosis was incorporated to have a minimum number of samples to perform the statistical analysis. The inclusion criteria were the following: women older than 18 years who consulted spontaneously due to abnormal vaginal discharge or who were clinically diagnosed at the time of the normal routine consultation in the examination. The exclusion criteria were: people under 18 years of age, women with antibiotic treatment during the last 30 days and/or who were taking immunosuppressive drugs or who have immunosuppression, women who were in their menstrual period at the time of taking the medication were also excluded sample and women who have had sexual intercourse in the last 48 h. Regarding the question Were these women symptomatic of vaginitis? Yes, they were all symptomatic, all the samples of women included in this study (except the control group) presented an abnormal vaginal discharge, presence of erythema in the vaginal walls and external genitalia, in addition to the manifestation of itching. These details were included in the material and methods section in the “Study population and ethical approval” section.
Q5: Please also include more details about the control group? Are these lab controls or women with no a etiological diagnosis? Please define in the methods how controls are defined.
A5: Thank you for this feedback. Control group samples come from archival plates from a previous project. They correspond to samples from healthy patients without abnormal vaginal discharge and with normal microbiota according to the Nugent criteria classification. Controls were defined as CA-negative and TV-negative (determined by PCR) and normal microbiota determined by Gram stain and applying Nugent's criteria for microbiota classification. These details were included in the material and methods section in the “Study population and ethical approval” section.
Reviewer 2 Report (New Reviewer)
It is interesting, but difficult to read manuscript. I think the authors have taken too many different vaginal conditions (TV, VVC, and BV) for analysis that making the results poorly grounded. Moreover, they analyzed samples of mixed or co-infection (Table 1B) when the impact of each infection agent is complicated to define. In summary, a very small number of participants with various vaginal conditions makes the results less reliable. The authors should analyze samples of one type of infection with the respective number of control samples.
Introduction & Discussion:
1. Typical BV is characterized by a lack of inflammation and the absence of leukocytes (Donders, 2007). Why BV condition is selected for NETs analysis?
2. Lines 307-308: “Vaginal discharge contains abundant PMN accompanying the various bacterial vaginoses”. It is not clear how the authors assume vaginitis and vaginosis.
Methods: the diagnosis of the forms of vaginitis (as the authors stated) has several inconsistencies and drawbacks detailed below:
1. Candida albicans is a part of the normal vaginal microbiota of healthy asymptomatic women (Sobel, 2007; Sobel et al. 2015). Detection by PCR C.albicans does not indicate candidiasis or VVC (Sobel et al. 2015). The authors did not indicate whether the C.albicans-positive women reported any characteristic symptoms of VVC. Conventional microscopic methods along with patient complaints are used to diagnose VVC. If the authors mix CA with VVC, they should clarify the issue. In lanes 222-223, the authors stated that “The samples diagnosed through PCR as CA infections presented pseudohyphae that were easily visualized using Gram staining”. They should clearly state in the methods section how the diagnosis of infection has been performed.
2. The authors did not indicate whether women participating in the study had any vaginal complaints. The presentation of Amsel criteria will strongly support the data of the Nugent scoring.
Results: Fig. 1 (B). Nugent criteria MI. Please, correct.
Author Response
Comments from Reviewer 2:
It is interesting, but difficult to read manuscript. I think the authors have taken too many different vaginal conditions (TV, VVC, and BV) for analysis that making the results poorly grounded. Moreover, they analyzed samples of mixed or co-infection (Table 1B) when the impact of each infection agent is complicated to define. In summary, a very small number of participants with various vaginal conditions makes the results less reliable. The authors should analyze samples of one type of infection with the respective number of control samples.
Introduction & Discussion:
Q1: Typical BV is characterized by a lack of inflammation and the absence of leukocytes (Donders, 2007). Why BV condition is selected for NETs analysis?
A1: Thanks for the comment. In unpublished data, we still performed a Gram stain reading and, in parallel, a PMN count using Giemsa stain. The origin of including the Giemsa stain analysis was due to the criteria used by Bacova (https://www.fba.org.ar/wp-content/uploads/2021/06/Atlas-Bacova-Erige-2018.pdf ) in which they report the inflammatory reaction in the flow test, and we have seen that BV with an inflammatory reaction (presence of neutrophils) in this scenario, we are facing BV as an acute or recent condition, because neutrophils appear in infections in the first 4 or 5 days. After this period, macrophages, lymphocytes and other leukocytes appear in the focus of infection, facing an event of chronic inflammation (10.1111/j.1600-0897.2010.00902.x).
Q2: Lines 307-308: “Vaginal discharge contains abundant PMN accompanying the various bacterial vaginoses”. It is not clear how the authors assume vaginitis and vaginosis.
A2: Thanks for the comment. To clarify this, literally Vaginitis is an inflammation of the vagina characterized by an abnormal vaginal discharge or secretion, itching, irritation, or burning, where a microorganism is not necessarily involved. Thus, non-infectious vaginitis can be caused by an allergic reaction to vaginal washes, soaps, condoms, among others, and infectious vaginitis where there may be the participation of various microorganisms (10.1097/AOG.0000000000002090). Bacterial vaginosis (produced by mainly anaerobic bacteria) has been described as the most frequent vaginitis followed by Candidiasis and Trichomoniasis (10.1016/j.ogc.2017.02.010). Following these definitions, we have used the concepts well throughout the manuscript.
Q3: Methods: the diagnosis of the forms of vaginitis (as the authors stated) has several inconsistencies and drawbacks detailed below:
Candida albicans is a part of the normal vaginal microbiota of healthy asymptomatic women (Sobel, 2007; Sobel et al. 2015). Detection by PCR C.albicans does not indicate candidiasis or VVC (Sobel et al. 2015). The authors did not indicate whether the C. albicans-positive women reported any characteristic symptoms of VVC. Conventional microscopic methods along with patient complaints are used to diagnose VVC. If the authors mix CA with VVC, they should clarify the issue. In lanes 222-223, the authors stated that “The samples diagnosed through PCR as CA infections presented pseudohyphae that were easily visualized using Gram staining”. They should clearly state in the methods section how the diagnosis of infection has been performed.
A3: We thank the reviewer for commenting on CA cases. In this regard, we can comment on the diagnosis of CA infection: a paragraph is introduced in the method section to clarify the choice of the 3 cases specified as CA. These cases were chosen according to the clinical diagnosis of Candidiasis and their representative clinical manifestations of a vulvovaginal Candidiasis and corroborated by a specific PCR for Candida albicans. As additional background information, we informed the reviewer of the clinical manifestations observed by the midwife during the speculoscopy: case (4) Presence of erythema and abrasions on the external genitalia and erythema on the vaginal walls and cervix. Moderate flow amount with thick appearance. In relation to the discomfort reported by the patient, these were: itching, burning and dyspareunia. Case (5) presented erythema on the external genitalia and erythema on the vaginal walls and cervix and had a moderate amount of discharge with a thick and lumpy appearance. The complaints reported for this case were: pruritus, burning, dysuria, and dyspareunia. And in case (6) she had erythema/congestion and red stippling of the cervix, the vaginal discharge she presented was moderate and lump-like in appearance, with symptoms of itching and dyspareunia.
Q4: 234: The authors did not indicate whether women participating in the study had any vaginal complaints. The presentation of Amsel criteria will strongly support the data of the Nugent scoring.
A4: Thanks for the comment. Under the conditions of our study where archive samples were used, it is not possible to present the Amsel criteria, since this evaluation was not considered from the beginning. However, we add details of the symptomatology presented by the women who were included in this study in the section "Study population and ethical approval".
Q5: Results: Fig. 1 (B). Nugent criteria MI. Please, correct.
A5: Thanks for noticing this error, we have corrected it in the new manuscript.
Round 2
Reviewer 2 Report (New Reviewer)
The authors haven't made substantial improvements to the manuscript. The issues remained to be addressed:
1. Figure 1: thus, a total of 10 samples including controls (n=3, nos 8-10?, Figure 1B) have been presented for the study. Among tested samples, only 1 represents “pure” BV (no. 7? Figure 1B) without concomitant infections tested by the described methods. Is one BV sample enough to state about PMN and NETs in this condition? Thus, the statements in lines 410-412 should be softened or omitted. It is complicated to perform any generalized conclusions (see Discussion) from the data of a single sample.
The same issue where the impact of the type of microbiota is studied (Fig. 5 D): too small number of samples and “pure” controls.
2. The presence of PMN in BV is still under debate https://pubmed.ncbi.nlm.nih.gov/15459411/ , https://pubmed.ncbi.nlm.nih.gov/31369673/ that should be discussed in the Introduction or Discussion sections. Unfortunately, not all reviewers are familiar with the Spanish that is written your cited pdf file. https://www.fba.org.ar/wp-content/uploads/2021/06/Atlas-Bacova-Erige-2018.pdf.
How do you explain that the PMN score in the control sample no. 8 coincides with “pure” BV sample no.7 (Fig. 1 C).
3. To clarify the detection of VVC/CA. I accept an explanation of observed clinical manifestations in the author's response, but the information should be clearly presented in Methods: CA was detected by 1) clinical signs 2) microscopic examination 3) the absence/presence of CA detected by 1+2 was confirmed by PCR.
4. Historically BV has been named “vaginosis” instead of “vaginitis” to emphasize the absence of inflammation as vaginitis (again historically) is linked with inflammation. I agree that BV is a sort of vaginitis, but “various bacterial vaginosis” (lines 332-333) should be corrected.
5. The authors should describe the limitations of their study.
6. The references linked with vaginal conditions (especially BV and PMN) should be revised.
7. G.vaginalis should be replaced by Gardnerella spp. (line 406)
Author Response
Thank you very much for your constructive comments.
Q1: Thank you very much for your constructive comments.Figure 1: thus, a total of 10 samples including controls (n=3, nos 8-10?, Figure 1B) have been presented for the study. Among tested samples, only 1 represents “pure” BV (no. 7? Figure 1B) without concomitant infections tested by the described methods. Is one BV sample enough to state about PMN and NETs in this condition? Thus, the statements in lines 410-412 should be softened or omitted. It is complicated to perform any generalized conclusions (see Discussion) from the data of a single sample.
The same issue where the impact of the type of microbiota is studied (Fig. 5 D): too small number of samples and “pure” controls.
A1: We understand the reviewer's comment that we cannot draw conclusions from a BV sample, but in this study by grouping the samples by microbiota status: 3 BV samples for our analyses, 1 CA positive sample, the second TV positive and the third without infection by these last two. To our knowledge, if the question we ask ourselves is whether the BV condition is different from IM or NM, in terms of what we are comparing, then our group variable is BV vs NM, independent of CA infection and TV present/absent, this would be valid according to other similar works where data are grouped by pathology independent of the pathogen that is causing it (https://doi.org/10.1007/s10815-020-01883-7). Then we wanted to see the effect of CA, TV and controls how they behave in terms of the presence of NETs and NETotic cells, independent of the state of the microbiota to visualize the effect of these infections, which is why were grouped by pathogen and presented the results as follows. Due to the reviewer's comments, we envision some limitations of our study which were detailed in the ' study limitations' section. (Line 444-450).
Lines 410-412 were modified according to the reviewer's suggestion. (Line 414-417).
Regarding the controls, we consider them valid due to the origin and the subsequent studies carried out on these samples in the laboratory. For greater clarity, these warn of women who attend the routine consultation for Papanicolaou smear collection and who presented clinically normal flows (physiological flows), the women did not have manifestations in the external genital (erythema, edema, others.), nor findings abnormal on speculoscopy (This information has been added to complement the description of the control groups in the new manuscript). Subsequently, in the laboratory, the vaginal smears of these cases were analyzed with Gram stain applying Nugent's criteria. The 3 cases were considered to have normal microbiota with a score between 0 and 3 and to be negative for CA and TV evaluated by specific PCR. We are aware that the number of cases studied is low, however, this research was conceived from the beginning as an exploratory study, mainly because other studies have been carried out in vitro and our study, unlike the previous ones, explores reality more closely. clinical samples obtained directly from routine processes in primary health care.
Finally, an important point that we would like to mention about our data (counts of NETs and NETotic cells) is that they were performed on many cells (because the automated program that was used allowed a detailed analysis of individual NETotic cells). This validates this exploratory study from our perspective. Undoubtedly, the future analysis of a greater number of samples and the inclusion of groups with a single infection and a group with co-infections will further enrich the results obtained.
Q2: The presence of PMN in BV is still under debate https://pubmed.ncbi.nlm.nih.gov/15459411/, https://pubmed.ncbi.nlm.nih.gov/31369673/ that should be discussed in the Introduction or Discussion sections. Unfortunately, not all reviewers are familiar with the Spanish that is written your cited pdf file. https://www.fba.org.ar/wp-content/uploads/2021/06/Atlas-Bacova-Erige-2018.pdf.
How do you explain that the PMN score in the control sample no. 8 coincides with “pure” BV sample no.7 (Fig. 1 C).
A2: Thank you for this comment. We agree that the presence of PMN in BV is controversial according to the research published so far, this has been added to the discussion section of the new version, considering the citations suggested by the reviewer (Line 411-412). We also apologize for sharing the link to the Bacova manual in Spanish. The English version of Bacova (guidelines for the diagnosis of vaginosis- vaginitis in primary care of women in fertile age or menopause) is available at the following link https://www.fba.org.ar/wp-content/uploads/2021/06/GUIDELINES-FOR-THE-DIAGNOSIS-OF-VAGINOSIS-VAGINITIS-IN-PRIMARY-CARE-OF-WOMEN-IN-FERTILE-AGE-OR-MENOPAUSE.pdf.
Regarding the comment about the similar presence of PMN in BV and the control samples without infections cannot ensure that they have a low amount of PMN, because the presence of PMN does not only depend on the presence/absence of infections, but also on many other factors such as the host's immune system (innate response), the person's lifestyle, diet, among others (DOI: 10.1002/wsbm.1458). On the other hand, the diagnosis of BV does not ensure a high amount of PMN as we have shown in Figure 1C and discussed during the new version of the manuscript (Line 411-412). Therefore, it is feasible to find a similar amount of PMN in VB and control group.
Q3: To clarify the detection of VVC/CA. I accept an explanation of observed clinical manifestations in the author's response, but the information should be clearly presented in Methods: CA was detected by 1) clinical signs 2) microscopic examination 3) the absence/presence of CA detected by 1+2 was confirmed by PCR.
A3: Thank you for the observation, we have added the information suggested by the reviewer in the new version of the manuscript. (Line 116-119)
Q4: Historically BV has been named “vaginosis” instead of “vaginitis” to emphasize the absence of inflammation as vaginitis (again historically) is linked with inflammation. I agree that BV is a sort of vaginitis, but “various bacterial vaginosis” (lines 332-333) should be corrected.
A4: Thank you for noticing this detail, we have corrected the sentence in the new manuscript. (Line 336-337)
Q5: The authors should describe the limitations of their study.
A5: Thank you for this recommendation. We have added the information suggested by the reviewer after the discussion, section “Study limitations”. (Line 445-450)
Q6: The references linked with vaginal conditions (especially BV and PMN) should be revised.
A6: the references have been revised, and we have added two citations indicating that the number of PMNs in BV is variable or still under debate. (Line 56-56, 411-412)
Q7: G.vaginalis should be replaced by Gardnerella spp. (line 406).
A7: Thank you for the observation made. The change has been made in the new version of the manuscript. (Line 409)
Round 3
Reviewer 2 Report (New Reviewer)
Final remarks:
1. Line 444: "a small-size population" should be replaced by "small number of clinical specimens (or samples)".
2. Line 335: "accompanying bacterial vaginoses" should be replaced by "accompanying vaginitis".
Author Response
Final remarks:
Q1: Line 444: "a small-size population" should be replaced by "small number of clinical specimens (or samples)".
A1: Thank you for this correction, we have made the change in the new version of the manuscript. (Line 447).
Q2: Line 335: "accompanying bacterial vaginoses" should be replaced by "accompanying vaginitis".
A2: Thank you for this correction, we have made the change in the new version of the manuscript. (Line 336).
This manuscript is a resubmission of an earlier submission. The following is a list of the peer review reports and author responses from that submission.
Round 1
Reviewer 1 Report
Conclusions of the article are based on 3 samples in each experimental group which seems low to sustain the case.
Vaginal Neutrophil numbers change during the phases of the ovarian cycle. Authors should take that account in the experimental design.
Every figure of the article should contain the experimental and control pictures.
Scale bar should be in the pictures.
Introduction and discussion are too long and confuse. It is not clear to the reader the conclusions of the article.
This article should be reviewed by a professional academic writer.
Reviewer 2 Report
- In TV infection, all three NETs are present, is there an overlap resulting in the inability to observe the single NET?
- If cross-infection exists, how do you identify which pathogen is responsible for the three NETs?
- ExceptSytox orange,could you find another dye for experiments?
- Is the mechanism of NET presentable in this study?
234: Why is the abbreviation here different from in figure 1B? NM in manuscript, but MN in figure1B.
240-241: Fluorescent pictures of the control group should also be placed in figure1D.
244: The result description uses figure2E,F...., but it uses e,f.....in the images,
245: Please specify blastoconidia and pseudohyphae with arrows
248-250: The phenomenon in the figure2f,i is not sufficient to prove this conclusion
267-268,273-274,277-278,281-283:There are no statistically significant differences,IM and BV ,CA and TV not presented in figures3 A,B,C,D.
265-283: This description has multiple duplicates and needs to be rewritten.
306-307: there showed no significant differences between the groups with normal, intermediate microbiota and bacterial vaginosis in sprNET. However, a significant difference between NM and BV in diffNETs. Compared with diffNETs, corresponding to the values on the left side of the histogram, the difference between NM and BV is observed to be greater in the sprNETs, and the data need to be reanalyzed statistically.
312-313: According to the description of 312-313, the diffNETs number should be definited in the vertical coordinate of Figure 5C,D
341-346: The relevant mechanism is not studied in this manuscript and can be removed from the discussion.
391-393:CA/TV-vaginitis presented at least three morphological phenotypes, such as diffNET, aggNET and sprNET, with a higher presence of aggNET in the case of TV-mediated vaginitis, Why higher is diffNETs in Figure 5A, the data does not seem to correspond between figure 4 and figures 5A.
397-402: Lack of similar presentation in the introduction.
406-410: Not relevant to this research and needs to be rediscussed.
445-448: “High presence of NETotic cells and neutrophil extracellular traps in vaginal discharges of women with vaginitis: an exploratory study”is the title of the manuscript, should be discussed around the title.
488,489,494,496 and so on, Change trichomonas vaginalis/candida albicans to trichomonas vaginalis/candida albicans in references
according to the journal's requirements, adding Funding. Institutional Review Board Statement, Data Availability Statement.
Figure1: Please annotate a and b in figure1A, the sample name for each lane should be clearly marked in the figure1 A. figures2D can not clearly see the phenomenon, please place a clear picture of 300dpi.
Figure2:the scale should be visible. specify blastoconidia and pseudohyphae with arrows in figure2a,b.
Figure3:The description of the fluorescence map of image 3 is not shown in the results.
Figure4: the scale is not visible and lack of clarity of image in figure 4,especially diffNETs,
Figure4: sprNETs, aggNETs and diffNETs in the left side of figure4
Figure5: the diffNETs number should be definited in the vertical coordinate of Figure 5C,D